# Analysis of Different Perspectives of Community-Based Long-Term Day-Care Centers

**DOI:** 10.3390/ijerph22071017

**Published:** 2025-06-27

**Authors:** Jui-Ying Hung, Pin-Hsuan Chiang, Kai-Lin Li

**Affiliations:** Department of Golden-Ager Industry Management, Chaoyang University of Technology, Taichung 413, Taiwan; s11235606@gm.cyut.edu.tw (P.-H.C.); s11235609@gm.cyut.edu.tw (K.-L.L.)

**Keywords:** community-based long-term day-care centers, evaluation, consensus benchmarks, management effectiveness

## Abstract

This study involved the secondary data analysis of empirical results obtained from the 2023–2024 on-site evaluation of 26 community-based long-term day-care centers (referred to as day-care centers) in Taiwan to determine the status of performance. Although the on-site evaluation results used in this study were all qualified (an evaluation score ≥70 points), a score range of 73.625–96.625 points were widely observed. The evaluation was carried out from the following perspectives: management effectiveness, professional care quality, safe environment, and the protection of individual rights. The results showed that management effectiveness was positively correlated with professional care quality, safe environment, and the protection of individual rights (*r* = 0.498~0.596); professional care quality was moderately correlated with safe environment and the protection of individual rights (*r* = 0.482~0.495); and safe environment and the protection of individual rights had the lowest correlation (*r* = 0.296). A paired-samples t-test showed that the average difference between each pair (institutional self-evaluation and evaluation committee score) in the consensus benchmark was significant. These findings suggest that the overall service quality and management efficiency of day-care centers can be improved by strengthening internal self-assessment training, improving consensus benchmark definitions, and upgrading management processes so that the evaluation results reflect the actual implementation status.

## 1. Introduction

According to the United Nations (UN) World Population Prospects (WPP) 2022, the global population of individuals aged over 65 is projected to reach 1.6 billion by the mid-21st century, and the population ratio will grow from 10% in 2022 to 16% by 2050. According to National Development Council (NDC) estimates, the elderly population accounts for 20.8% of the total population in 2025, and Taiwan is entering a super-aged society, defined as “the first year of super-aged era (it meant the population aged over 65 accounts for 21% of the total population)” [1]. As the population structure continues to move towards super-aged, the number of older people with dementia or disabilities continues to increase; in Taiwan, the population of people with disabilities is expected to increase by approximately 18% to 20% every five years. To address this challenge, the Ministry of Health and Welfare (MOHW) (2023) launched the “10-year long-term care plan 2.0 (2016–2025)” (referred to as LTC 2.0) in 2016 and will continue with the “10-year long-term care plan 3.0 (2026–2035)” (referred to as LTC 3.0) in 2026 [2]. Dai (2025) asserts that the impact of LTC 3.0 on day-care centers mainly manifests in three ways: (1) the diversified transformation of service models, (2) resource integration and community connection, and (3) flexible services and individualized care [3].

The LTC 2.0 policy actively promoted a diversified LTC service system, especially community-based day-care centers, to help alleviate the pressure of family care in Taiwan. Day-care centers provide a place for daily activities and professional care for people with disabilities and dementia. This is an example of a highly acceptable LTC service model in which older people receive proper care during the day, reducing the pressure on family caregivers, and then return to a familiar environment (mainly home) at night. The LTC industry tends to lack manpower, but the group care approach used in day-care centers requires fewer staff members (one caregiver is responsible for 6–10 elderly people). According to the statistics of the Department of LTC of the MOHW, the number of day-care centers increased from around 400 in 2019 to 730 by the end of 2024, with day-care centers being located in 89.5% of the secondary school districts in Taiwan (MOHW, 2024a) [4].

In 2018, the government in Taiwan fully promoted LTC 2.0 and developed community LTC services at three levels: A, B, and C. The extensive distribution of LTC services under the LTC 2.0 policy helps to alleviate social isolation, promote social participation, enhance the sense of choice and control, and prevent and delay disability. The levels are defined as follows. Level A: Community Integrated Service Center. When individuals or their families are in need of LTC services and make requests, the care management specialists complete an assessment and then assign a case manager from Level A to draft a care plan. This plan is supported by a multidisciplinary professional care team that provides customized and personalized support services. Level B: Complex Service Centers. These include home-based, community-based (divided into day-care, small-scale multifunctional care, group homes, and family care), and comprehensive care. Level C: Community LTC Station. This is the LTC provision most deeply integrated into various corners of the community, serving primarily healthy or sub-healthy elderly individuals or those in the early stages of dementia or disability. It provides opportunities for communal dining, social activities, and social participation alongside other clients, and health-promotion activities to prevent and delay the onset of disability. In 2024, there were 4033 LTC service units in Taiwan, with the most common being home-based services (2235 units, 55.42%) and the least common being institutional accommodation (122 units, 3.03%). Community services, which include day-care, family care, group homes, and small-scale multifunctional care, total 1478 units (36.65%), with day-care facilities accounting for the highest number (951 providers, 64.34%) and group homes the lowest (34 providers, 2.3%) (Figure 1). Although the types and services of Level B LTC centers are different, they still have some common elements, such as serving people with disabilities and dementia [5]. The service content incorporates social participation, physical activity, and goal-oriented support, such as advice and guidance, life or dementia care, and frailty reduction, to enhance or maintain health in place.

According to Article 39 of the Law on LTC Services, the main unit responsible for the public sector shall provide guidance to LTC institutions alongside supervision, assessment, inspection, and evaluation. In addition, according to Article 3 of the Measures for the Evaluation of LTC Service Institutions, the sponsoring authority is responsible for the evaluation of community-based LTC institutions and home-based LTC institutions in the county (or city) [5].

As a key link within the LTC system, day-care centers are of great significance in the care of older people with LTC needs, reducing family burden. However, with the increase in the number and coverage rate of day-care centers, improving the operation management efficiency and service quality of these centers has become a challenge. The main purpose of this study is to explore the self-evaluation and committee evaluation of day-care centers and to put forward practical suggestions for improving their quality of service.

## 2. Related Literature

### 2.1. Long-Term Care (LTC) Policy

The development of LTC policy in Taiwan began in the 1980s, with family and non-governmental volunteer services comprising the main resources in the early period. However, LTC resources relying on unregistered institutional nursing services were limited. With the acceleration of medical progress and population aging, the government has gradually attached more importance to the needs of elderly people. The LTC system, a social welfare and service system, was established to cope with the aging population and the increase in the number of people with disabilities and dementia. The 10-year plan for LTC 2.0 (2022) refers to the provision of continuous, multilevel support and care for older people and those with disabilities. The system covers the integration of medical care, life support, community services, and family care resources, and emphasizes the core concept of “aging-in-place” [6,7].

In addition to actively developing LTC services at home, in the community, and in institutions, LTC 2.0 also includes a new payment system and establishes an overall community care service. The government is also committed to improving the salaries of care workers, stabilizing human resources, and increasing the coverage of dementia services to meet the needs of various types of LTC service users and comprehensively improve the accessibility and quality of these services [8]. LTC 2.0 services are provided in four main areas: (1) care and professional services; (2) transportation services; (3) assistive device rental and barrier-free space repair services; and (4) respite services. Since 2018, the MOHW has adjusted the payment standards used in LTC 1.0, adopted a parcel payment model, introduced LTC benefits and payment standards, and further revised subsidy standards and costs to reduce the financial burden on family members and provide multiple care options. Community-based institutional services mainly include non-accommodation-based day-care centers and non-24 h care support. Day-care centers have always been a relatively important part of the infrastructure for promoting community care in Taiwan, which is not only the focus of LTC 2.0 but also the endpoint of formal care services in the community [9]. In addition, day-care centers are mainly based on a fixed-point model, usually for older people or those who have temporary care needs in the community, providing care services at institutions during the day with the service users returning home at night to maintain interaction with family members; this approach helps to reduce psychological decline [10]. Thus, day-care centers are often regarded as having the characteristic effect of deinstitutionalization.

In terms of the evolution of LTC systems and policies, the Integrated Community Care System in Japan serves as a global model. It connects medical care, LTC, and community support through nursing insurance and has built a local support network based on the four levels of self-help, mutual aid, collective assistance, and public aid to meet the demand imposed by the aging “baby boomer” generation, particularly those aged over 75 [11]. In Singapore, the adoption of the model has strengthened respite services, and qualitative research shows that providing short-term respite care can significantly alleviate the burden on family caregivers and enhance their caregiving sustainability [12]. South Korea has also incorporated user satisfaction and interdepartmental coordination indicators into its evaluation system to improve the applicability of assessments and policy responsiveness. To address the shortcomings of current evaluations regarding user satisfaction and end-of-life care, the authors of [13] suggest introducing a “person-centered” evaluation tool, focusing on the quality of interactions between caregivers and care recipients, participation in decision-making, and living environments. In addition, regarding the practicality and diversity of evaluation indicators, the research has introduced 43 evaluation items, covering operations, environmental safety, rights protection, and service processes, through a comparison of Korea and Japan, and proposed a four-level evaluation process to improve efficiency [14,15].

Finally, innovative technology provides new opportunities for upgrading the quality of LTC. Experimental studies of telepresence robots have shown that they can significantly reduce emotional isolation for family caregivers while simultaneously enhancing interaction and emotional support for residents [16]. Overall, multinational empirical experience proves that combining regulatory systems, performance management, talent cultivation, community integration, and technology is essential to create a sustainable, user-centered, high-quality LTC system.

### 2.2. The LTC Centers Evaluation System

The term “evaluation” is often used interchangeably with “assessment”, “appraisal”, and “judgment”, and has similar meanings and uses [10]. The evaluation of LTC centers is a statutory mechanism of the government and is the core of the systematic evaluation of the quality of LTC service providers. The main purpose of this evaluation is to ensure that the care services provided meet certain quality standards to protect the quality of LTC, personal rights and interests, and the health and safety of individuals. The evaluation system also promotes the continuous improvement of the internal management of the centers, enhances the professional capacity of care, and guides the implementation of high-performance resource allocation and high-quality performance in day-care centers. The LTC Services Act was enforced on 3 June 2017, and according to Article 39, local governments shall provide guidance to LTC institutions, alongside supervision, assessment, inspection, and evaluation. Evaluation is one of the key ways to ensure the quality of LTC service institutions [17].

The phenomenon of super-aged is observable across the globe; as such, the quality of LTC centers has become a focus in both policy-making and practical management. Numerous studies have explored how to construct a structured and person-centered care system from various perspectives. In the monitoring and improvement of the quality of LTC services, the relevant research emphasizes how residents’ subjective experiences should be the core focus. These experiences include aspects such as autonomy, social relationships, participation in daily activities, environmental compatibility, and dietary services, all of which work together to create a quality of life that truly meets the needs of elderly people [18]. Moreover, the researchers in [19] explain how the government, third-party evaluation agencies, and users communicate through the systematic collection of quality information and the adoption of public mechanisms for evaluating the results. The researchers recommend regularly reviewing the structure, process, and outcome indicators and publicly disclosing the evaluation results to enhance market transparency in Taiwan. Regarding the construction of evaluation criteria, process and outcome aspects have gradually been incorporated into early structural monitoring, and user satisfaction is included to achieve dynamic tracking feedback with the aim of continuously improving and transforming care-quality monitoring indicators in long-term care institutions [20]. In terms of management tools for LTC institutions, the Balanced Scorecard (BSC) was adopted and combined with the Analytic Hierarchy Process (AHP) to construct performance indicators for day-care centers in Taitung County. It has also been proposed that “learning and growth” and “customer satisfaction” are the most critical success factors for the operation of LTC institutions, which is in agreement with the user-centered LTC service model [21].

The care system in the UK is centered on community care as an integral part of the national health care policy, and quality management is implemented through the Care Standards Act [22]. To ensure the quality of the services provided by care providers in the UK, the Care Quality Commission (CQC) utilizes on-site evaluations and other inspections. On-site evaluation includes three forms: (1) regular inspections, which are comprehensive inspections undertaken without notice once a year; (2) reactive inspections, which are unannounced inspections that can be conducted an unlimited number of times for special events or for no specific reason; and (3) thematic examinations, which focus on specific topics, such as medical management, nutrition, or services for people with disabilities. In 2004, the Ministry of Health, Labor and Welfare in Japan issued the Guidelines for Third-Party Evaluation of Welfare Services, which clearly state that the purpose of third-party evaluation is to help organizations understand and improve management issues. To ensure fairness and professionalism, the evaluation includes the assessment of the basic policy of welfare services, the organization and operation of the services, and their implementation. The evaluation results and suggestions for improvement are declared with the consent of the organization [23]. South Korea implemented Long-Term Care Insurance (LTCI) in 2008, establishing a unified demand assessment and institutional evaluation system, insured by the National Health Insurance Corporation, across the country. The core of this system is a nationally standardized functional assessment that classifies beneficiaries into six levels (Level 1 to Level 6) to provide home and institutional services according to different needs. Every two years, public insurance units are mandated to conduct quality inspections of institutions, and this, combined with a multilevel evaluation system—from self-assessment and third-party evaluation to regular supervision by the central government—ensures that the assessment indicators consider structure, process, and outcome [24]. The assessment indicators include organizational operation, environmental safety, and client rights/service effectiveness, and a payment method based on daily, hourly, or itemized charges. The transparency of the public evaluation results strengthens user choice (a comparison of evaluation implementation models in various countries is presented in Table 1).

The evaluation system focuses on structural and normative management, but lacks user satisfaction in Taiwan. In Japan, it emphasizes the neutrality and transparency of third-party institutions, and the system is carefully designed. In the UK, user participation and the frequency of on-site inspections are particularly prominent, and the management of operators is the strictest and most transparent. In South Korea, the assessment mechanism of LTCI includes standardized assessments, regular supervision, and a multilevel, multi-entity assessment network, forming a quality assurance system that combines fairness and transparency. Further, it continuously optimizes user participation, client satisfaction indicators, and interdepartmental coordination to implement high-quality, people-centered LTC.

In terms of human resource training, establishing a clear functional grading and modular training system effectively enhances the professional skills and retention rates of care service workers. Taking the Eternal Trust Social Welfare Foundation in Taiwan as an example, its three-stage functional grading of basic → advanced → specialized, combined with on-the-job rotation and supervision mechanisms, significantly strengthens team self-discipline and mutual trust (Huang, 2021) [25]. In addition, the physical and mental health of employees must be taken seriously, as reports show that Canadian caregivers face stress from balancing paid work and unpaid family-caregiving, which needs to be alleviated through flexible working hours and respite support policies [26].

## 3. Research Methodology

### 3.1. Research Design

The evaluation standards of community-based day-care centers are the reference version published by the central government. The mayoral care centers of each county adjust the status of LTC centers and the provision of LTC services within their jurisdiction and announce this information at the end of the year, before the on-site evaluation. The evaluation benchmarks include 10 items for management effectiveness (code: A), 12 items for professional care quality (code: B), 13 items for safe environment (code: C), and 5 items for the protection of individual rights (code: D). Each concept has its own benchmark and description. The maximum score is 100 points, and there are 40 consensus benchmarks for day-care centers, each of which is worth 2.5 points. Each consensus benchmark score is further classified into five levels: A+ (2.5), A (2.125), B+ (1.75), B (0.875), and C (0). The passing score for LTC centers is 70, so the B+ (1.75) score is designed to be 70% of A+ (2.5). The sources of evaluation include the main managers of the day-care center (defined as the internal evaluation score) and the on-site evaluation committee (defined as the external evaluation score). The internal evaluation scores must be submitted to the relevant authority before the on-site evaluation and then passed on to the evaluation committee for review [27].

### 3.2. Data Collection and Measurement

This study utilizes a secondary data analysis of the empirical results from the 2023–2024 on-site evaluation of 26 community-based day-care centers in one county. Due to the increasing demand for LTC services and the promotion of the policy “One junior high school district, One Day-care center”, the achievement rate for day-care centers in 2025 is expected to reach 90.0%. This study is mainly based on secondary data obtained from the on-site evaluation of an LTC service-quality improvement project involving day-care centers in a target county. To date, 32 school districts in the target county have met the “One junior high school, One Day-care center” policy requirements (an achievement rate of 94.11%). Therefore, this study focuses on 26 day-care centers in the target county (26 school districts, covering 81.25% of the total) as the research subjects. According to the LTC Services Act and the Evaluation Methods for LTC Service Institutions, day-care centers are required to be evaluated every four years. If the center does not meet the standards, it may be re-evaluated the following year. A total of four evaluators are responsible for each on-site evaluation: one from administrative management, two from LTC and nursing, and one from the environmental safety category. Regarding expert validity, during the consensus meeting for all evaluation committees, detailed explanations are provided regarding the evaluation goals, structure, benchmark, and meanings through discussions.

In this study, on-site evaluations were conducted of 26 community-based day-care centers in a specific county from 1 August 2023 to 31 July 2024. A score of 70 or above was considered a pass, and all of the centers met this threshold (a pass rate of 100%).

### 3.3. Data Analysis

SPSS 18 (PASW Statistics Base) was used for descriptive statistical analysis, and Pearson correlation analysis and paired-sample *t*-tests were used as data analysis methods to examine the cognitive differences between the internal evaluations of day-care centers and the external scores from the evaluation committee. The average score for the 26 day-care centers is 85.574 (1.125), with scores ranging from 73.625 to 96.625 (the scores for the benchmark concepts are shown in Table 1). Due to the wide score range, standard error was used instead of standard deviation (S.D.). Table 2 shows the average values of the consensus benchmarks for community-based day-care centers in various concepts. There are 10 consensus benchmarks for management effectiveness, with each consensus benchmark scoring 2.5 points. The actual average score for the management effectiveness concept is 20.98 points (with a full score of 25 points), resulting in a compliance rate of 20.98/25 = 83.92%. Therefore, the evaluation results for the community-based day-care centers show that the compliance rate for the protection of individual rights is the highest, at 91.16%, followed by safe environment, at 87.13%, management effectiveness, at 83.92%, and, lastly, professional care quality, at 82.94%.

## 4. Results

Pearson correlation analysis was used to explore the intrinsic correlation between the self-assessment of the centers and the assessment made by the evaluation committee across four concepts in order to understand the differences and consistencies between internal perceptions and external assessments in service quality evaluation. A further investigation was made of the importance of the evaluation system for day-care centers in terms of practical applications and suggestions for improvement.

### 4.1. Pearson Correlation Analysis Results for Evaluation Committee Scores Across the Four Main Concepts

Table 3 presents the correlation analysis of the four concepts of the evaluation committee. It can be observed that management effectiveness is moderately highly positively correlated with professional care quality, safe environment, and the protection of individual rights (r = 0.498 to 0.596). Professional care quality shows a moderate correlation with safe environment and the protection of individual rights (r = 0.482 to 0.495), while safe environment has no significant correlation with the protection of individual rights (r = 0.296).

### 4.2. Paired-Sample t-Test: Differences in Consensus Benchmarks Between Centers and Evaluator Perspectives

From the paired-sample t-test, the test results of the consensus benchmark for the four concepts A, B, C, and D (Table 4) indicate that, for most consensus benchmarks, the scores from the self-assessments made by the center (internal score) are significantly higher than those provided by the evaluation committee (external score) (except for eight benchmarks—B7, B10, B11, C1, C7, C8, D1, and D3—which showed no significant differences).

## 5. Discussion

In this study, we conducted an in-depth investigation of the different evaluation scores of 26 day-care centers, analyzing the differences and interrelationships between the scores given through self-evaluation, as well as those provided by the evaluation committee, across four concepts and forty consensus benchmarks. The results are as follows.

The average scores and S.D. for each concept show that self-evaluation generally results in higher scores than the evaluations made by the committee. This indicates that day-care centers have a significantly positive perception of their own operations. The finding reflects that day-care centers evaluate themselves with a more optimistic attitude, while external committees conduct on-site evaluations based on their professional expertise and the scoring benchmark discussed in consensus meetings, which often leads to conflicting evaluations.

A moderately positive correlation exists among the various day-care center concepts, with a low correlation only being found between safe environment and the protection of individual rights. This indicates that there are certain differences or diverging perceptions among evaluators regarding the consensus benchmarks and on-site observations. This cognitive gap should remind managers and decision-makers to review the existing self-evaluation system and consensus benchmarks. In the future, internal self-evaluation training should be strengthened, the definitions of consensus benchmarks should be explained, and management processes should be enhanced to ensure that the evaluation results better reflect the actual implementation status. In this way, the overall service quality and management efficiency of day-care centers can be continuously improved [14,15].

## 6. Conclusions

### 6.1. Professional Care Quality and Safe Environment Have a Significant Positive Impact on Management Effectiveness

According to the results, the day-care center self-assessment model uses professional care quality and safe environment as explanatory variables, both of which have a significant positive impact on management effectiveness. This indicates that day-care centers perform well in terms of cross-professional services, health management, and safety equipment maintenance, and the overall management effectiveness is also relatively good. This result emphasizes that optimizing care quality and improving safety environments are both key concepts in enhancing the management effectiveness of day-care centers [27].

### 6.2. Protection of Individual Rights and Interests Has Limited Impact on Evaluation Results and Management Effectiveness

When evaluating day-care centers, whether through self-assessment or the scoring model used by the evaluation committee, the performance regarding the protection of individual rights does not significantly impact the results. This may reflect the fact that the consensus benchmark for this concept is predominantly based on statutory regulations and contracts, which are essential baseline benchmarks for operating a day-care center. Therefore, the difference in operational effectiveness influenced by the protection of individual rights is not significant. However, if there is no consensus benchmark corresponding to this concept, the day-care center may be unable to continue providing services under special agreements with the relevant authorities [13,26].

Day-care centers should focus on enhancing management effectiveness in two major ways: professional care quality and safe environment. Moreover, it is important to be aware of the gap between internal self-assessment and external evaluation in order to establish a more objective and transparent assessment and improvement system.

### 6.3. The Government and Relevant Agencies Should Provide More Resources and Guidance

The relevant authority can further strengthen the guidance and supervision of day- care centers by providing policy subsidies, resource integration, and regular training activities to help enhance management efficiency and service quality. In addition, promoting inter-agency information exchange and experience-sharing, forming a positive cycle of progress, will also help narrow the cognitive gap between internal and external evaluations to enhance the overall quality of services [12,16,19,20,21,22,23].

## 7. Limitations and Future Research Directions

### 7.1. Limitations of Sample Range and Representativeness

This study only involved evaluation data from 26 day-care centers in a specific county. While the results might reflect the current situation in that region, they cannot comprehensively represent the circumstances of other regions or organizations of different scales. Future research could expand the sample sources to include more counties or nationwide day-care centers in order to enhance the external validity and representativeness of the research results.

### 7.2. Limitations of Cross-Sectional Data

Cross-sectional data analysis was used in this study, an approach that cannot reveal changes in the management and service quality improvements of each day-care center over time. In the future, it is recommended that a longitudinal tracking research design be adopted, regularly collecting evaluation data to explore the long-term effects of improvement measures on management performance. A comprehensive understanding of the sustained benefits of improvement strategies could then be obtained.

### 7.3. Subjectivity and Insufficient Quantification of Evaluation Criteria

Some evaluation criteria may be biased due to the reliance on respondents’ subjective assessments, leading to significant discrepancies between internal and external ratings. Future research might consider designing objective quantitative data, such as actual service operation records, user satisfaction survey data, and third-party objective evaluation indicators, to supplement the deficiencies of self-assessment and committee ratings, thereby allowing the evaluation results to more accurately reflect service quality.

### 7.4. External Environmental Variables Are Not Sufficiently Incorporated

The operational management efficiency of day-care centers is influenced not only by internal management factors but also potentially by external policies, community resources, and demographic structures, among other factors. This study did not include external variables. Future research could incorporate external environmental factors to construct a multilevel impact model to further explore the moderating effects of external resources and policy environments on organizational performance.

## Figures and Tables

**Figure 1 ijerph-22-01017-f001:**
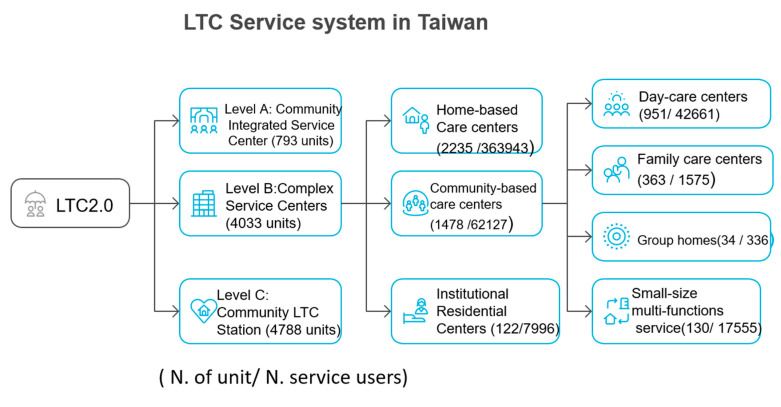
LTC service system in Taiwan (2024).

**Table 1 ijerph-22-01017-t001:** Comparison of evaluation implementation models in various countries.

	Taiwan	Japan	South Korea	UK
Supervisory Authority	MOHW: institutional accommodation Local Government: home-based and community-based LTC centers	Third-party evaluation unit, certified and supervised by the prefectural government	National Health Insurance Corporation	The CQC, an independent agency established by the UK government
Evaluation Mode	Once every four years On-site evaluation Appointing members from professional fields	Third-party organizations evaluate through written materials, interviews, and on-site evaluation	Every two years, public insurance units are mandated to conduct quality inspections of institutions; this is combined with a multilevel evaluation system that uses self-assessment, third-party evaluation, and regular supervision by the central government	Regular, irregular, and professional on-site evaluations; interviews with patients and family members
Method of Disclosing Evaluation Results	Evaluation results announced by government, but lack of subjective satisfaction	Evaluation results are completely open and transparent, available for public reference	Evaluation results are made public to help users make decisions	Published online within 14 days after evaluation, updated once a month
Evaluation Concepts	Management effectiveness, professional care quality, safe environment, and protection of individual rights	Service philosophy of the organization, operational management, individual service quality for users	Organizational operation, environmental safety, client rights/service effectiveness. Payment method based on daily, hourly, or itemized charges	Health and personal care, daily life, complaint handling, environmental equipment, personnel management

**Table 2 ijerph-22-01017-t002:** The consensus benchmark scores for day-care LTC service centers.

Main Concept	*n* of Consensus Benchmarks	Full Scores of Benchmarks/ Each	Average Scores of Main Concepts from Evaluation Committee		Full Points of Main Concept	Total Scores of Each Concept from Evaluation Committee		Coincidence Rate (%)	Order of Coincidence Rate ^1^
			Internal	External		Internal	External		
Management effectiveness (A)	10	2.5	2.2736	2.098	25	22.736	20.98	83.92	3
Professional care quality (B)	12	2.5	2.271	2.073	30	27.25	24.881	82.94	4
Safe environment (C)	13	2.5	2.37	2.178	32.5	30.828	28.318	87.13	2
Protection of individual rights (D)	5	2.5	2.382	2.279	12.5	11.909	11.395	91.16	1
*n* of service centers (26 units)	20	40	2.318	2.139	100	92.723	85.574	85.574	

^1^ The compliance rate ranking reflects the ratio between the actual total score and the full score for each main concept. A higher rank (closer to the top) indicates a higher level of compliance. For example, the concept ranked first is the most compliant.

**Table 3 ijerph-22-01017-t003:** Pearson correlations among evaluators’ perspectives for the four concepts.

Pearson Correlation Analysis *p*	Management Effectiveness	Professional Care Quality	Safe Environment	Protection of Individual Rights
Management effectiveness Pearson correlation analysis *p*	1	0.595 ** 0.001	0.596 ** 0.001	0.498 ** 0.010
Professional care quality Pearson correlation analysis *p*	0.595 ** 0.001	1	0.482 * 0.013	0.495 * 0.010
Safe environment Pearson correlation analysis *p*	0.596 ** 0.001	0.482 * 0.013	1	0.296 0.142
Protection of individual rights Pearson correlation analysis *p*	0.498 ** 0.010	0.495 * 0.010	0.296 0.142	1

* *p* ≤ 0.05; ** *p* ≤ 0.01.

**Table 4 ijerph-22-01017-t004:** Analysis of the differences between different perspectives in the consensus benchmark.

Difference Score in Consensus Benchmarks (Internal, External)	Pairwise Differences			*t*	*p*
	μ	s	S.D. Average		
A1 Business plan development and implementation	0.235577	0.316873	0.062144	3.791	0.001 ***
A2 Social participation and community resource linkage status	0.129808	0.210597	0.041301	3.143	0.004 **
A3 Administrative operations and service quality management	0.173077	0.218165	0.042786	4.045	0.000 ***
A4 Disclosure of service information	0.216346	0.216673	0.042493	5.091	0.000 ***
A5 Financial management system	0.115385	0.231633	0.045427	2.540	0.018 *
A6 Deficiencies and improvement evaluation by the relevant authority during the period of supervision/inspection	0.144231	0.238988	0.046869	3.077	0.005 **
A7 Supervisor actually participates in administrative and care quality management meetings/activities	0.225962	0.377524	0.074038	3.052	0.005 **
A8 Establishment and implementation of systems related to staff rights and interests	0.235577	0.298594	0.058559	4.023	0.000 ***
A9 Regular health check-ups and follow-ups for staff	0.086538	0.161126	0.031599	2.739	0.011 *
A10 Pre-training for new staff	0.192308	0.359219	0.070449	2.730	0.011 *
B1 Service plans and multi-professional services	0.245192	0.414549	0.081300	3.016	0.006 **
B2 Client adaptation counseling or support	0.216346	0.241241	0.047311	4.573	0.000 ***
B3 Infection prevention, treatment, and monitoring during service delivery	0.269231	0.484371	0.094993	2.834	0.009 **
B4 Health check-ups and health management for service users	0.173077	0.285212	0.055935	3.094	0.005 **
B5 Handling and prevention of accidents and emergencies	0.250000	0.425735	0.083493	2.994	0.006 **
B6 Emergency medical evacuation services are available	0.192308	0.430451	0.084418	2.278	0.032 *
B7 Handling of group or community activities for service recipients	0.100962	0.271614	0.053268	1.895	0.070
B8 Supportive services provided for caregivers (related persons)	0.225962	0.413853	0.081163	2.784	0.010 **
B9 Living assistance for client	0.254808	0.419048	0.082182	3.101	0.005 **
B10 Maintain self-care skills	0.149038	0.441615	0.086608	1.721	0.098
B11 Strengthen LTC service personnel	0.100962	0.310281	0.060851	1.659	0.110
B12 Nutritious meal service	0.192308	0.403471	0.079127	2.430	0.023 *
C1 Rest equipment provided	0.072115	0.184300	0.036144	1.995	0.057
C2 Daily activities provided	0.144231	0.282162	0.055337	2.606	0.015 *
C3 Clean and hygienic kitchen and dining environment	0.259615	0.295641	0.057980	4.478	0.000 ***
C4 Emergency call system	0.235577	0.316873	0.062144	3.791	0.001 ***
C5 Food hygiene	0.288462	0.594526	0.116596	2.474	0.021 *
C6 Fire safety management	0.177885	0.374455	0.073437	2.422	0.023 *
C7 Public safety inspection of building	0.048077	0.362417	0.071076	0.676	0.505
C8 Evacuation system (evacuation setup)	0.057692	0.229757	0.045059	1.280	0.212
C9 Formulate and implement emergency disaster response plans and operating procedures that meet the characteristics and needs of day-care LTC institutions	0.346154	0.548074	0.107486	3.220	0.004 **
C10 Institutional environmental cleanliness and vector control	0.259615	0.331517	0.065016	3.993	0.001 ***
C11 Equipment maintenance and management	0.264423	0.361121	0.070822	3.734	0.001 ***
C12 First aid items	0.177885	0.337589	0.066207	2.687	0.013 *
C13 Safe and clean drinking water supply	0.177885	0.314589	0.061696	2.883	0.008 **
D1 LTC institution security insurance	−0.028846	0.147087	0.028846	−1.000	0.327
D2 Service contract with the client or family member	0.129808	0.258556	0.050707	2.560	0.017 *
D3 Fees and receipts	0.057692	0.229757	0.045059	1.280	0.212
D4 Establish and handle the feedback response/grievance process	0.177885	0.337589	0.066207	2.687	0.013 *
D5 Service satisfaction surveys	0.177885	0.255375	0.050083	3.552	0.002 **

* *p* ≤ 0.05; ** *p* ≤ 0.01; *** *p* < 0.001. The gray background indicates no significant difference.

## Data Availability

Statistical data are included in the article. No data sets were generated or analyzed during this study.

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
