# Peer review of "Analysis of Different Perspectives of Community-Based Long-Term Day-Care Centers"

_ijerph, 2025, doi:10.3390/ijerph22071017_

Round 1
Reviewer 1 Report
Comments and Suggestions for Authors
Thank you for the opportunity to review this interesting manuscript.
Overall the paper needs to be improved in all areas identified above. The English needs to be improved to enhance overall clarity of ideas and content. It is an interesting topic. I assume that evaluation here is a formal accreditation type process for the programs. Perhaps that can be stated more specifically.
A few considerations.
- define "super-agers"
- check English and grammar throughout the paper
- Can you elaborate on who did the scoring and how? E.g. how were the internal scores determined? by whom?
- For Table 2 - suggest labelling the full variable name rather than "A" or "B"
- In Table 3 - which are the external and internal scores - not clear from how these are labelled
- There is no reference to obtaining REB approval for this work.
- The authors have not discussed limitations of the work and/or future work. For example, addition of qualitative methods could further explore areas where gaps were noted in the quantitative data.
- Please link your discussion back to other literature - how are your findings similar or different?
The topic is an important one - I think an interesting contribution to the literature.
Comments on the Quality of English Language
English needs to be improved to enhance clarity.
Author Response
Please see the attachment, thanks!

Reviewer 2 Report
Comments and Suggestions for Authors
Summary: The study shows a secondary data analysis of the empirical results from the 2023–2024 on-site evaluation of 26 community-based day care centers in Taiwan. The focus of the study is on the comparison between self-evaluation and evaluation by external evaluation committees regarding: (A) Management Effectiveness, (B) Professional Care Quality, (C) Safe Environment, and (D) Protection of Individual Rights. The study applies SPSS for descriptive statistical analysis, Pearson correlation, and paired-sample t-tests. The paper's key contribution is its analysis of self-evaluations and external evaluations in Taiwanese day care centers. The results of the paper offer basis for international comparison of long-term care (LTC) services. Therefore, paper's findings represent an advancement of knowledge in the field of LTC.
General concept comments: The research question is relevant and the hypotheses are scientifically valid. However, the introductory section and the literature review require improvement. Authour should expand and more contextualize literature review, especially with the results from comparative LTC systems (only UK is mentioned but it is not explained why only UK was selected).
Detailed comments:
The hypotheses are sound and the methodology adheres to established empirical standards. However, the paper lacks a clear rationale for the selection of the 26 day care centers from a total of 730 in Taiwan. More detail is needed to justify the sampling.
The introduction lacks a comprehensive overview of Taiwan’s LTC system. It briefly refers to three types of care (A, B, and C) but fails to define or describe these types.
The role of day care centers within "Level B" services should be more clearly articulated. Relevant statistics on the proportion of older adults engaged in day care centres in comparison with the ones using other LTC services in Taiwan should be added (the paper reveales the number of centers but omits the number of individuals utilizing these services, which is critical for assessing their impact).
The concept of “self-evaluation« must be described more thoroughly, including how these evaluations are conducted and validated. While evaluation benchmarks are outlined, the text of the article should offer greater detail and explanation of the 10 items in A (Management Effectiveness), 12 in B (Professional Care Quality), 13 in C (Safe Environment), and 5 in D (Protection of Individual Rights), rather than relying solely on tables.
The literature cited is sparse. I recommend adding at least 20–30 more sources, especially those focusing on LTC regulation in Asian countries, to enhance the regional and international relevance of the article.
The article explaines LTC evaluation practices in the UK, but without proper referencing or clear justification for selecting the UK for comparison The authors should select at least three comparative systems—preferably including two from Asia—in addition to the UK, and explain the rationale behind their selection. Furthermore, at least five similar empirical studies from other countries should be included and critically compared.
Key Criteria:
Novelty: The paper addresses an original question and contributes new data to the understanding of LTC services in Taiwan. However, international comparative analysis is necessary to enhance its broader relevance.
Scope: The article aligns well with the scope of the journal.
Significance: The results are appropriately interpreted and are significant for evaluating community-based LTC services.
Quality: The data are appropriately analyzed, but the presentation can be improved through clearer narrative and expanded discussion.
Scientific Soundness: The study is technically sound, with appropriate use of statistical methods. The tools and procedures are generally described well.
Reader Interest: The study will be of interest for researchers in the LTC field, especially those focused on quality of day-care centres, comparative evaluation, and regional policy development.
Overall Merit: The paper contributes knowledge in LTC systems and addresses a relevant issue of evaluation of LTC services. It is scientifically valid, but it would benefit from expanded context, and richer literature.
Comments on the Quality of English Language
English language can be improveed.
Author Response
Please see the attachment, thanks!

Round 2
Reviewer 1 Report
Comments and Suggestions for Authors
The authors have addressed the concerns I had with the previous version satisfactorily.
The manuscript is much improved.
Reviewer 2 Report
Comments and Suggestions for Authors
My prevoius suggestions were observed by authors, therefore I suggest the paper to be accepted in the proposed form.